# Do Drinking Motives Mediate the Relationship between Neighborhood Characteristics and Alcohol Use among Adolescents?

**DOI:** 10.3390/ijerph16050853

**Published:** 2019-03-08

**Authors:** Gina Martin, Joanna Inchley, Candace Currie

**Affiliations:** 1Child and Adolescent Health Research Unit, School of Medicine, University of St Andrews, St Andrews KY16 9TF, UK; Joanna.Inchley@glasgow.ac.uk (J.I.); cec53@st-andrews.ac.uk (C.C.); 2Human Environments Analysis Laboratory, Western University, London, ON N6A 3K7, Canada; 3Department of Geography, Western University, London, ON N6A 3K7, Canada; 4MRC/CSO Social and Public Health Sciences Unit, University of Glasgow, Glasgow G2 3AX, UK

**Keywords:** neighborhood, deprivation, drinking motives, adolescence, mediation, multilevel, urban, rural, social cohesion, disorder

## Abstract

Adolescents not only vary in their alcohol use behavior but also in their motivations for drinking. Young people living in different neighborhoods may drink for different reasons. The aims of this study were to determine if neighborhood characteristics were associated with adolescent drinking motives, and whether drinking motives mediate the relationship between neighborhood context and regular alcohol use. Data from the Scottish Health Behaviours in School-aged Children 2010 survey of students in their 4th year of secondary school were used. The study included 1119 participants who had data on neighborhood characteristics and had used alcohol in the past year. Students were asked questions about the local area where they lived, their alcohol use, and their motives for drinking alcohol, based on the Drinking Motives Questionnaire Revised Short Form (DMQR-SF). Multilevel multivariable models and structural equation models were used in this study. Coping motives showed significant variation across neighborhoods. Structural equation models showed coping motives mediated the relationships between neighborhood deprivation, living in an accessible small-town, and neighborhood-level disorder with regular alcohol use. Public health policies that improve neighborhood conditions and develop adaptive strategies, aimed at improving alcohol-free methods for young people to cope better with life’s stresses, may be particularly effective in reducing inequalities in adolescent alcohol use if targeted at small towns and areas of increased deprivation.

## 1. Introduction

Research has shown that adolescent alcohol consumption varies across neighborhoods [1,2]. Adolescents also vary in their motivations for drinking [3]. The Scottish Government’s Alcohol Framework identifies that it is crucial to understand motives for drinking to limit the negative impact of alcohol on individuals and society [4]. Additionally, gaining such information is vital for the design of effective public health strategies [3,5,6,7].

Drinking motives are often regarded as the final pathway to alcohol use, which link to various drinking patterns and may mediate more distal influences [7,8]. Currently there are no studies that examine the extent to which adolescent drinking motives vary across neighborhoods; existing studies have examined perceptions of neighborhoods but ignore neighborhood membership. Understanding if drinking motives are associated with where adolescents reside allows for a better comprehension of the pathways by which neighborhood impacts adolescent alcohol use. Therefore, exploring the associations of neighborhood conditions with drinking motives has been identified as an important area for research [9].

Drinking motives are defined as the reasons why people drink, with the assumption that people drink to obtain a desired outcome [10]. These motives can be conceptualized as representing dimensions of a motivational construct [8]. Using this model, the motives to drink are categorized by two underlying dimensions which were proposed by Cox and Klinger [11]: valence (positive or negative forces that attract or detract i.e., pleasantness or utility [12]) and source (internal or external) of the outcomes individuals expect to achieve from alcohol use. In terms of valence, it is theorized that people drink to gain positive outcomes or to avoid negative consequences. In terms of source, internal motives of “enhancement of a desired internal emotional state or by external rewards such as social approval or acceptance” ([6] p. 11) also underlie drinking behavior. Four drinking motives are generally recognized. These are drinking for: (1) coping, (2) enhancement, (3) social and (4) conformity motives [7]. These four motives are commonly measured using a four factor Drinking Motives Questionnaire, known as the DMQR (Drinking Motives Questionnaire Revised) which was developed by Cooper [10] and has been validated for use in several samples (adults, university students and adolescents) [13,14,15].

The dimensions and source map onto the four motives as follows:Internally generated, positive reinforcement = enhancement, i.e., drinking to have fun and get drunkExternally generated, positive reinforcement = social, i.e., to better enjoy social gatheringsInternally generated, negative reinforcement = coping, i.e., to alleviate problems and worriesExternally generated, negative reinforcement = conformity, i.e., not to feel left out

Motives are often theorized as a potential pathway between neighborhood characteristics and alcohol use. The most common hypothesis is that in stress-inducing neighborhoods (i.e., neighborhoods experiencing material deprivation and disorder) alcohol is used to cope with the increased pressure that comes from living in such an environment [16,17,18]. Previous research has shown that a greater frequency of stressful events occurs in low-income neighborhoods [19]. Additionally, adverse neighborhood conditions may reduce adolescents’ psychological coping resources; this may lead to increased substance use to deal with life’s challenges [20]. In contrast, in areas where norms are in favor of alcohol, it would be expected that extrinsic motivations (i.e., social and conformity motives) would be higher. Despite these theorized relationships and suggestive evidence, there are few studies that have examined the motivational pathways through which neighborhoods may impact alcohol use among the adolescent population.

A review undertaken by Kuntsche et al. [13] examined factors related to adolescents’ alcohol use motives and found that sex, age, mental state (i.e., depression), and situation (i.e., drinking at a party) were all related to motives to drink. The review only found macro-level factors, as measured by cross-national differences in sociocultural factors, in the literature; studies examining within-country neighborhood differences were not identified [13]. However, as posited above, the neighborhood characteristics that adolescents are exposed to may impact on their drinking motives. A study of Portuguese adolescents examined whether adolescents’ self-reported perceptions of their neighborhoods were associated with their motives for drinking. This study found that drinking for enhancement, to be social, to conform, and to cope, were higher when adolescents perceived high levels of night-time entertainment, violence and robberies, and when they reported that they live in an isolated area. However, perceived social cohesion in their local area was not associated with any drinking motives [5]. Research that examined associations of drinking to cope with family and individual characteristics found that adolescents from higher socio-economic family backgrounds drank more to increase confidence; while those from families from lower socio-economic backgrounds drank more to cope with low mood [3]. Although these studies suggest there may be an association between neighborhood characteristics and adolescent drinking motives, they deal only with individual perceptions of the neighborhood or family background, and so little is known about whether the external observable conditions of where adolescents live are associated with their drinking motives. A study of US adults found that objectively measured neighborhood disadvantage was negatively correlated with social motives for drinking and positively correlated with drinking to cope (to forget worries and problems) [9]. However, whether these relationships exist among adolescents is unknown.

Research is needed to identify sub-groups of young people who have varying drinking motives [8]. Additionally, calls have been made for further research to investigate drinking motives as a potential mediator in the relationships between neighborhood characteristics and alcohol outcomes [9]. Accordingly, this study will examine associations between social and physical dimensions of the neighborhood; and adolescent drinking motives. The potential for drinking motives to mediate the relationship between neighborhoods and alcohol use is also explored.

Research questions:Are characteristics of the neighborhood associated with adolescent drinking motives?If so, is there evidence that motives mediate the relationship between neighborhood conditions and adolescent alcohol use?

## 2. Materials and Methods

### 2.1. Sample

Survey data were collected as part of the 2010 Scottish Health Behaviour in School-aged Children (HBSC) survey, a World Health Organisation cross-national study [21]. The 2010 Scottish survey of pupils in S4 (the fourth year of secondary school: typically 15 years old), included a boosted sample of non-urban schools to allow for examination of a range of urban-to-rural classifications (total sample n = 3591) [22]. Ethical approval for the 2010 Scottish HBSC survey was granted by the University of Edinburgh’s School of Education Ethics Committee.

Residential location was determined by self-report of home postcode by the pupils. Ethical approval for use of the postcode data, which is not part of the public data file, was granted by the University of St Andrews Teaching and Research Ethics Committee (reference number: MD11023). In Scotland, postcodes represent a small geographical area making it possible to geocode (assign a latitude and longitude) based on this information. Additionally, many administrative boundaries are built of postcodes; for example, Scottish data zones (DZ) and intermediate data zones (IDZs) are higher levels of geography which contain multiple postcodes. DZs (of which there are 6505) have on average 750 residents. IDZs are built up from DZ, representing 1235 regions in Scotland, containing on average 4000 residents. IDZs were developed based on administrative data and local knowledge [23]. Administrative data available at these higher levels of geography allowed for linkage of alcohol outlet densities (AOD), urban/rural classification and neighborhood deprivation to the HBSC survey data [2].

To be included in the analysis students had to report their home postcode and reside in an IDZ that had 5 or more survey participants (n = 1558). This allowed for adequately reliable measures of neighborhood-level social cohesion and disorder [2,24,25]. From there, students were included in the analysis only if they reported positively to having ever drank and to having consumed alcohol in the past 12 months (n = 1119), as the drinking motives questions were only asked of current drinkers. Those included in the study were significantly (p<0.05) more likely to be in the high family-affluence tertile (40 versus 34%) than those excluded but were not significantly different in terms of family structure and sex.

### 2.2. Measures

#### 2.2.1. Drinking Motives

Drinking motives were measured using the four factor Drinking Motives Questionnaire Revised Short Form (DMQR-SF). This questionnaire was developed and validated by Kuntsche and Kuntsche [26] from the larger DMQR, developed by Cooper [10]. In the short form version each of four dimensions (coping, enhancement, conformity, and social) are measured using the average of three items assessed on a 5-point Likert scale (almost) never/some of the time/about half of the time/most of the time/(almost) always. This questionnaire has been previously validated for use with adolescents [27]. Conformity motives were measured with the following questions: In the last 12 months, how often did you drink… (1) To fit in with a group you like? (2) To be liked? (3) So you won’t feel left out? Coping motives were measured with the following questions: In the last 12 months, how often did you drink… (1) Because it helps you when you feel depressed or nervous? (2) To cheer up when you’re in a bad mood? (3) To forget about your problems? Enhancement motives were measured with the following questions: In the last 12 months, how often did you drink… (1) Because you like the feeling? (2) To get high? (3) Because it’s fun? Social motives were measured with the following questions: In the last 12 months, how often did you drink… (1) Because it helps you enjoy a party? (2) Because it makes social gatherings more fun? (3) Because it improves parties and celebrations?

Imputations were carried out in cases where one item was missing on a motives scale, using the person-scale average (enhancement = 45 cases (4%); social = 20 cases (1.8%); conformity = 6 cases (0.5%); coping = 4 cases (0.4%) [28]. Cronbach’s alphas show good reliability, as follows: enhancement = 0.798, social = 0.925; conformity = 0.854, coping = 0.898.

All motives were log-transformed to reduce skew and approximate a normal distribution. This was done after preliminary examination found that the residuals in the full models were not normally distributed. Descriptive statistics are reported on the motives before transformation.

#### 2.2.2. Alcohol Use

Weekly drinking was determined from the question: “at present how often do you drink anything alcoholic, such as beer, wine, or spirits? Try to include even those times when you only drank a small amount” Responses included frequency of consumption (every day, every week, every month, hardly ever, and never). Those who reported drinking any beverage type daily or weekly were coded as weekly drinkers.

#### 2.2.3. Neighborhood Measures

Neighborhood deprivation was determined by the income domain of the Scottish Index of Multiple Deprivation (SIMD) 2012, which is comprised of several indicators of income deprivation [29]. This measure was based on quintiles of all DZs in Scotland [22,30]. The two most deprived categories were combined, as few of the sample (<10%) resided in the most deprived quintile. Urban/rurality was classified into 6 categories based on classifications by the Scottish Government: (a) large cities (population of 125,000 or more), (b) other urban (population >=10,000 and <125,000), (c) accessible towns (population 3000–9999 and within a 30 min. drive to a settlement >=10,000), (d) remote towns (population 3000–9999 and more than a 30 min. drive to a settlement of >=10,000, (e) accessible rural (population <3000 and within a 30 min. drive to a settlement of >=10,000), and (f) remote rural (population <3000 and more than a 30 min. drive to a settlement >=10,000).

Data on alcohol outlet density were obtained from the Centre for Research on Environment, Society and Health at the University of Edinburgh who geocoded all premises that held a license to sell alcohol in 2012 based on postcodes. These data were used to estimate a measure of AOD using Kernel Density Estimation (KDE). This process divided Scotland into 100 × 100 m grid cells and assessed the number and proximity of outlets at a radius of the center of each cell. Outlets nearer the center were given greater weight than those further away; therefore, the value represents a proximity-weighted estimate of the density of each outlet type. Data were classified as on-trade (i.e., bars or restaurants) and off-trade (i.e., shops) [31].

Perceived social cohesion was measured using three questions: in the area where you live (1) you can trust people around here, (2) people say “hello” and talk to each other in the streets, and (3) it is safe for younger children to play outside. Responses ranged from “agree a lot” to “disagree a lot”, on a five-point scale. The Cronbach’s alpha of the sample at the individual level (perceived social cohesion) was 0.754. Perceived disorder was measured using the same procedure. Three questions were used in this measure: in the area where you live are there (1) groups of young people who cause trouble? (2) litter, broken glass, or rubbish lying around? and (3) run-down houses or buildings? Responses ranged from “none” to “lots”, on a three-point scale. The Cronbach’s alpha at the individual level (perceived disorder) was 0.758.

Neighborhood-level social cohesion and neighborhood-level disorder measures were constructed by combining individual responses within an IDZ unit. This was done using a three-level item response model (level 1 = item; level 2 = pupil; level 3 = neighborhood) accounting for item severity and pupil’s sex as an individual-level covariate. More details can be found in Martin et al. [24]. The reliability at the neighborhood-level for the social cohesion and disorder measures were 0.577 and 0.563, respectively. Previous research has considered similar neighborhood-level reliability values acceptable [32].

#### 2.2.4. Covariates

Sex was included as a covariate as previous studies have found differences between boys and girls on several of the drinking motives [33]. To test the influence of neighborhood deprivation independent of family socio-economic status [34], and because there is evidence that drinking to cope varies by family socio-economic condition [3], family affluence was included as a covariate. This was measured by a composite scale [35,36] using responses to four questions regarding family vehicle and computer ownership, having one’s own bedroom, and family holidays. The items were combined using categorical principal components analysis to create tertiles of low, medium, and high family affluence, for the total sample [37]. Family structure was also included (living in a family with (1) both parents, (2) a single parent, or (3) a step-parent or other family composition) as there is some indication that the family environment may influence drinking motives [38,39].

Age was not included as a covariate as all the adolescents were in the same grade and there is no hypothesized reason for drinking motives to vary due to small differences in age (correlations between age and the drinking motives were all non-significant p> 0.05).

### 2.3. Statistical Analysis

Analysis was conducted in two stages (1) examining motives as an outcome, and (2) exploring for mediation (indirect effect) of motives on alcohol use.

In the first stage, associations between neighborhood characteristics and drinking motives were assessed using multilevel regression modelling. Empty models were tested to examine the variation of drinking motives across neighborhoods (IDZs). A linear model controlling for demographics and family characteristics (sex, family affluence, and family structure) (Model 1) was conducted. Neighborhood characteristics were then included in a subsequent model (Model 2). Individual perceptions were adjusted for in a final model (Model 3). These analyses were conducted using runmlwin in Stata and MLWin using Markov chain Monte Carlo (MCMC) methods. These procedures do not produce point estimates, instead many iterations are run and for each evaluation a distribution is formed. From this accuracy interval estimates are produced, namely credible intervals, which can be interpreted similarly to confidence intervals. The mean of the distribution can be obtained and used as substitute for a point estimate. A p value can be derived and interpreted as the probability of the (null) hypothesis [40]. A start value needs to be given for MCMC sampling. For this research, the values were given using a least squares method [41]. Generally, the model iterations take a while to ‘settle down’ (converge) so some iterations are omitted from the sample from which the summary values are drawn. This is called the burn-in [42]. Like past research, a burn-in length of 5000 was used [43]. The number of iterations (chain length) needed to achieve model convergence was set at 200 000 based on the Raftery-Lewis statistic and visual examination of the trajectory plots [43] (see Appendix A). Bayesian Deviance Information Criteria (DIC) was used to test model fit across models with a smaller DIC indicating better fit and a difference of 5 considered substantial [44,45].

Assumptions for using a linear regression model were evaluated by visual inspection of plots of residuals versus predicted values to examine for heteroskedasticity, residual Q-norm plots on the full models to test for normality, and a Global Moran’s I was calculated on the model residuals at the neighborhood-level to test for spatial autocorrelation in the model.

In the second stage of analysis, based on findings of the previous models (where neighborhood characteristics were associated with specific motives), potential mediation of the drinking motives on the relationship between neighborhood characteristics and alcohol use were explored. Cooper et al. [7] lay out the conditions necessary for mediation of drinking motives to exist on more distal influences of alcohol use. These include establishing that the independent effect (in this case neighborhood characteristics) predicts both motives and the alcohol use outcome. Furthermore, assessment of an indirect effect (mediation) was conducted using Mplus. Multivariable path analysis examined the potential mediating pathways of drinking motives on the relationship between neighborhood exposures and alcohol use. As recommended, a range of measures were used to assess model fit: comparative fit index (CFI) >0.95, root mean-square error of approximation (RMSEA) <0.06, and Tucker Lewis index (TLI) >0.95 [46]. Analysis was conducted using the COMPLEX sub-command to account for clustering by neighborhood (IDZ) [46]. The MODEL INDIRECT sub-command was used to estimate indirect effects and their standard errors [9]. The weighted least squares means and variance (WLSMV) estimator in Mplus was used as this is a robust estimator that does not assume normality, allows for fit indices and indirect effects, and as the model contained both continuous and categorical variables [9]. This estimator is based on probit regression using an inverse normal link function for categorical outcomes and linear regression for continuous outcomes. Paths controlled for all demographic and neighborhood exposure variables in the first instance, but to find the most parsimonious model and preserve degrees of freedom, paths on variables where *p*
>0.10 were removed [9]. Results are reported as unstandardized coefficients with standard errors (SE).

## 3. Results

The characteristics of the study sample are given in Table 1. The mean age of the sample was 15.6 and 49% of the sample were male. Social motives were the most commonly reported drinking motive (mean = 3.1; SE = 0.04), followed by enhancement (mean = 2.4; SE = 0.03), coping (mean = 1.7; SE = 0.03), and conformity (mean = 1.4; SE = 0.02).

### 3.1. Empty Models

Table 2 shows the results of the empty models. These models revealed that coping motives varied significantly by IDZ (improved Bayesian DIC when including the neighborhood-level, and z-score test was significant (*p* = 0.038)) with 5.2% of variation being explained by the neighborhood in which the adolescents resided. A cross-classified model was also specified for coping motives accounting for school-level variance, which reduced the amount of variance accounted for by neighborhood to 2.9% and Bayesian DIC improved (1627.80 versus 1633.47). Enhancement motives had 1.9% of variation explained by the neighborhood in which the adolescents resided and a very small improvement in Bayesian DIC was found when including the neighborhood-level (difference = 0.48). The other two drinking motives had <1.5% of variation explained by neighborhood.

### 3.2. Multivariable Models

Although empty models supported that coping and, to some degree, enhancement motives were the only motives to vary across neighborhoods, further models were still conducted on all four motives to examine the associations with demographic factors and perceived neighborhood conditions, as previous work has found that perceptions of neighborhoods are predictive of adolescent drinking motives [5].

Table 3 gives the results of multivariable models. In models not adjusted for neighborhood characteristics (Model 1), males had lower coping motives than females, and those from single-parent families had higher coping motives compared to those from two parent families (Table 3, Model 1). In fully adjusted models (Table 3, Model 3) males had lower coping motives (β = −0.17, *p* = <0.001) but family structure was no longer significant (*p*
>0.05). Coping motives are approximately 16% lower in males than females, based on the geometric mean. Residing in an accessible small-town was positively associated with coping motives (β = 0.14, *p* = 0.048, approximately 15% higher) compared to those in urban regions. Additionally, those residing in the least income deprived areas (β = −0.16, *p* = 0.003, approximately 15% lower) and the less income deprived areas (β = −0.14, *p* = 0.005, approximately 15% lower) had lower coping motives, compared to those in the most deprived areas. Those in the third category of deprivation also had reduced coping motives but this finding only neared significance (*p* = 0.055) when accounting for individual perceptions. Neighborhood-level disorder was positively associated with coping motives, but when adjusting for perceptions of the neighborhood this was no longer significant (Table 3, Models 2 and 3). Perceptions of disorder were positively associated with coping motives (β = 0.03, *p* = 0.035). A 1 unit increase in perceived disorder was associated with about a 3% increase in coping motives. Including school-level variance by specifying a cross-classified model made little difference to the results (coefficients and p–value did not vary substantially) (see Appendix A). Although coping motives varied by neighborhood, the majority of variance is explained at the individual level; this is not unexpected given that drinking motives have been found to be related to intrinsic individual characteristics such as personality type [27].

For social enhancement and coping motives there were few instances where neighborhood characteristics were associated with motives (see Appendix A). When examining social motives for drinking, only perceived disorder was significantly associated with social motives (β = 0.03, *p* = 0.030). Perceived disorder was also positively associated with enhancement motives (β = 0.03, *p* = 0.039). Additionally, enhancement was positively associated with family structure, with students from single-parent families having greater enhancement motives compared to those living in two biological parent families (β = 0.10, *p* = 0.021, approximately 11% higher). Conformity motives were higher for males than females (β = 0.05, *p* = 0.033, approximately 5% higher) and lower in the second most income deprived neighborhood category compared to those in most deprived (β = −0.10, *p* = 0.010).

In terms of residual diagnostics, Moran’s I showed no significant autocorrelation between neighborhood residuals indicating that spatial autocorrelation is not of concern in these models. The Q-norm plots (see Appendix A) show that the individual-level residuals are still somewhat skewed despite log-transforming the motives. However, linear regression approaches tend to be robust in terms of the normality assumption, unless using the model to predict specific data points [47].

### 3.3. Path Analysis

Path analysis was carried out examining coping as a potential mediator because of the significant variation across neighborhoods and the observed associations between neighborhood conditions. Neighborhood deprivation and living in an accessible small-town have previously been found to be associated with regular alcohol use among Scottish adolescents [2]; therefore, analysis was conducted whereby a direct and indirect path model was specified as follows: deprivation → coping → weekly drinking and living in an accessible small-town → coping → weekly drinking. Additionally, as perceptions of disorder may explain the relationship between neighborhood-level disorder and coping motives an indirect pathway was specified as follows: neighborhood-level disorder → perceived disorder → coping → weekly drinking. The hypothesized pathways are depicted in Figure 1. Originally, correlations were specified between neighborhood-level disorder and neighborhood deprivation as well as neighborhood-level disorder and urban/rurality, but these were excluded in the final model as inclusion had a negative impact on model fit. A direct effect was not included from neighborhood-level disorder to weekly alcohol use based on previous findings [2]. Models controlled for family structure and sex. Family affluence was excluded as its inclusion had a negative effect on model fit and it was not significantly associated with any outcome variable in the model.

Table 4 shows the results from the path analysis. As hypothesized, neighborhood-level disorder was associated with individual perceived disorder (β = 5.09, *p*<0.001). Additionally, coping motives to drink were associated with weekly drinking (β = 0.66, *p*
<0.001). When accounting for coping motives, neighborhood deprivation and living in an accessible small-town were not significantly associated with weekly drinking (*p*
>0.05). Figure 2 shows the significant paths, with non-significant paths from the hypothesized model removed. CFI and RMSEA indicated good model fit was achieved; however, TLI equaled 0.92, not meeting the cut-off of 0.95.

### 3.4. Indirect Effects

No direct effects were found from neighborhood deprivation on weekly alcohol consumption when accounting for coping motives (*p*
>0.05 for all categories of deprivation). However, an indirect effect was present for those residing in the least deprived areas (β = −0.11, SE = 0.04, *p* = 0.002) and the second most deprived category (β = −0.09, SE = 0.03, *p* = 0.002) through coping motives to weekly drinking. The effect was also indirect for those living in an accessible small-town (β = −0.10, SE = 0.05, *p* = 0.037) through coping motives to weekly drinking. Moreover, significant indirect effects were found from neighborhood-level disorder through perceptions (β = 0.30, SE = 0.13, *p* = 0.022) and through coping motives (β = 0.12, SE = 0.3, *p*
<0.001).

## 4. Discussion

In line with past research, we found Scottish adolescents reported alcohol use most frequently for social motives, followed by enhancement, coping, and conformity [5,8,27]. Sex differences were only found for negative valence motives. Differences in coping motives are consistent with previous research from Cooper [10] who found that girls scored higher than boys in coping motives in early adolescence (13–15 years) but among older adolescents the reverse was found. We also found that boys had higher conformity motives, which is consistent with previous studies [27]. Sex differences in drinking motives are largely thought to be due to differences in personality traits with adolescent females typically being more anxiety sensitive than males and males being more extroverted and impulsive [27].

This study aimed to determine whether neighborhood conditions were associated with drinking motives and whether drinking motives mediate the link between neighborhood conditions and drinking outcomes among adolescents. There was little variation evident in social enhancement and conformity motives by neighborhood. Previous work with adults found that neighborhood affluence was associated with social motives [9]; however, such associations were not present for neighborhood deprivation in this sample of Scottish adolescents. Social motives for drinking are frequently reported among adolescents, across different cross-cultural contexts, and appear to be equally important across various neighborhood conditions. This finding supports the need for universal intervention strategies which provide social activities for young people that may be an alternative to alcohol use.

Considering the lack of variation across neighborhoods in social, enhancement and conformity motives, it is not surprising that few neighborhood characteristics were associated with these motives. One exception is that those in the second most deprived neighborhood category had lower conformity motives than those in the most deprived areas. This may be because those residing in the most deprived neighborhoods may be more susceptible to peer group pressures [48] and that pressure to conform to drinking practices is related in a non-linear fashion to neighborhood deprivation. Based on these findings there is little evidence that neighborhood conditions impact on adolescents’ positive valence, and the impact is also limited for conformity motives.

Coping motives were higher among those living in more deprived areas. This is in line with Karriker-Jaffe et al.’s [9] findings that adults in disadvantaged neighborhoods report more coping motives for alcohol use. The higher levels of coping motives are of particular concern in deprived neighborhoods, where stress levels may be high and coping resources limited [9]. Moreover, those who drink to avoid negative outcomes typically experience more negative use-related consequences [7]. Coping motives in adolescence have also been associated with problem drinking later in life [49]. Therefore, based on our findings, Scottish adolescents residing in more deprived areas or accessible small towns may be at greater risk of alcohol related harms which has implications for their immediate and longer-term health.

The link between deprivation and drinking to cope may be explained by two, not mutually exclusive, hypotheses: (1) deprived neighborhoods create stress due to the physical and social conditions of the neighborhood, and alcohol is used to cope with the stress created by the environment, and (2) those residing in such neighborhoods have different strategies for coping with life’s general stresses and are more likely to use alcohol to deal with problems [9]. Because coping-motivated drinking may represent a form of self-medication [3], our findings suggest the need to develop and evaluate strategies that can help individuals cope with negative affect without alcohol. Additionally, removing neighborhood stress through programs to improve neighborhood conditions may reduce drinking to cope, as well as having wider benefits.

Neighborhood-level disorder was associated with coping motives indirectly through perceived disorder. Moreover, perceived disorder was associated with social and enhancement motives. This is similar to findings that perceptions of violence and robberies, as an indicator of disorder, are associated with drinking to cope [5], highlighting that perceiving the local area as a more problematic neighborhood gives rise to stronger drinking motives generally. It is difficult to explain these relationships as they may result from an unmeasured confounding variable such as a personality trait; or it may be that observing disordered neighborhood conditions leads to a higher motivated state to drink alcohol. Future work is needed to disentangle these associations.

Those living in accessible small towns had higher coping motives than their peers living in large cities. Few studies have examined the health of adolescents residing in small towns on the periphery of larger urban areas. Research in the US examined affluent suburban adolescents compared to disadvantaged urban adolescents and found that suburban youth reported significantly higher levels of substance use than urban adolescents and that anxiety levels were also higher [50]. However, Scottish small towns that are located near urban areas may differ substantially from affluent US suburban areas; these small-town regions are seldom examined in research and more studies are needed to understand the health behaviors of adolescents in these areas. Adolescents in such areas are often overlooked in research compared to their urban and rural peers.

Mediation analysis found that neighborhood deprivation and living in an accessible small-town were indirectly associated with weekly drinking through coping motives. This supports previous research that found the effects of neighborhood socio-economic status on substance use outcomes were likely to be indirect [51,52]. Additionally, as discussed above, neighborhood-level disorder had an indirect relationship with weekly drinking through perceived disorder and coping motives, further highlighting that distal exposures are often transmitted through several links in a chain [53].

Drinking motives are a concept that may aid in better targeting and designing prevention and intervention programs for at-risk adolescents [54,55]. The current coping measures are based on drinking to deal with negative emotions, depression, anxiety, and low mood. Simply reducing access to alcohol for adolescents in neighborhoods at greater risk may impact on consumption but does not get to the root of why adolescents are drinking more frequently in these contexts. Therefore, public health strategies that also address the fundamental factors that lead to drinking to cope may be more effective at reducing geographic inequalities in adolescent health than those which focus solely on consumption.

This research has several strengths. First, it examines multiple conditions of the neighborhood to determine which specific conditions may be associated with adolescent drinking motives. Second, to the best of our knowledge, this is the first study which tests for a potential mediation effect of drinking motives on the relationship between neighborhood characteristics and alcohol consumption among adolescents. However, there are several limitations to consider. This work did not account for personality traits and so further work is needed to understand how personality might impact the relationship between neighborhood conditions and alcohol use. However, altering individual intrinsic factors such as personality may be more difficult than modifying or targeting the neighborhoods where young people reside. Furthermore, this study is cross-sectional, so causation cannot be inferred. Time-series analyses and evaluation studies are needed to understand the impact of changes in the neighborhood on drinking motives and alcohol use [56]. The data on alcohol outlet density and SIMD were from two years after the HBSC data collection. Moreover, the neighborhood definitions used in this study were based on administrative boundaries, which may not align with adolescents’ perceptions of their local area.

In conclusion, of the four motives examined, only coping motives varied significantly across neighborhoods. Based on these findings, public health policies that develop adaptive strategies to improve alcohol-free methods for young people to cope better with life’s stresses may be particularly effective at reducing inequalities, if targeted at young people living in accessible small towns or areas of high neighborhood deprivation. Additionally, reduction of coping motives and regular alcohol use may be a potential side effect of improving neighborhood conditions.

## Figures and Tables

**Figure 1 ijerph-16-00853-f001:**
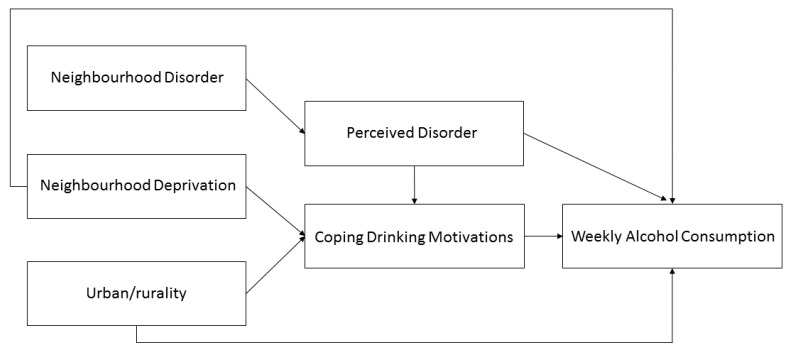
Hypothesized path model of neighborhood conditions on Scottish adolescent weekly alcohol consumption.

**Figure 2 ijerph-16-00853-f002:**
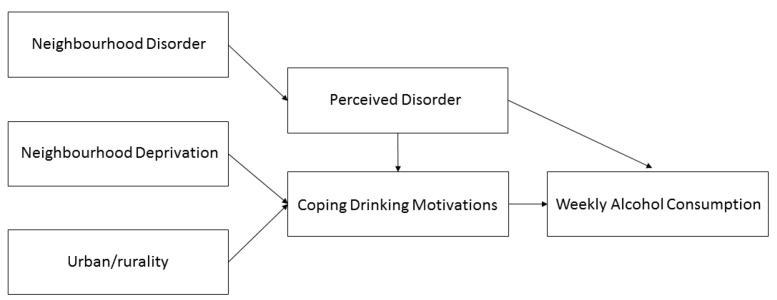
Path model of neighborhood conditions on Scottish adolescent weekly alcohol consumption with only significant paths present.

**Table 1 ijerph-16-00853-t001:** Sample characteristics (n = 1119).

Characteristics	Valid n	Mean (SE)/n (%)
**Demographics**		
Sex (male)	1119	545 (49%)
Age	1116	15.6 (0.01)
*Family affluence*	1119	
low		332 (30%)
medium		345 (31%)
high		442 (40%)
*Family structure*	1084	
both parents		760 (70%)
single parents		200 (19%)
other		124 (11%)
**Alcohol use and motives**		
Weekly drinkers	1119	391 (35%)
*Drinking motives*		
social	1111	3.1 (0.04)
conformity	1108	1.4 (0.02)
enhancement	1105	2.4 (0.03)
coping	1106	1.7 (0.03)
**Neighborhood perceptions**		
Perceived social cohesion	1095	12.0 (0.08)
Perceived disorder	1094	5.0 (0.05)
**Neighborhood characteristics**		
Alcohol outlet density (on-site)	1118	2.8 (0.13)
Alcohol outlet density (off-site)	1118	1.5 (0.05)
*Neighborhood deprivation*	1119	
1 (most deprived)		239 (21%)
2		260 (23%)
3		339 (30%)
4 (least deprived)		281 (25%)
*Urban/rural status*	1115	
large cities		149 (13%)
other cities		174 (16%)
accessible small-town		141 (13%)
accessible rural		188 (17%)
remote small-town		153 (14%)
remote rural		310 (28%)
Neighborhood-level social cohesion	1088	0.04 (0.01)
Neighborhood-level disorder	1079	−0.01 (0.00)

SE = standard error; Some percentages may not total 100 because of rounding.

**Table 2 ijerph-16-00853-t002:** Empty models of variation in drinking motives across neighborhoods (Intermediate Data Zones) (95% credible intervals).

	Social	Enhancement	Coping	Conformity
Neighborhood-level variance	0.004 (0.000, 0.012)	0.005 (0.001, 0.014)	0.014 * (0.002, 0.028)	0.002 (0.000, 0.006)
Individual-level variance	0.254 (0.233, 0.277)	0.242 (0.221, 0.264)	0.246 (0.224, 0.270)	0.156 (0.144, 0.170)
Neighborhood% of variance accounted for	1.3%	1.9%	5.2%	1.2%
Improvement in Bayesian DIC with neighborhood inclusion	No	Yes	Yes	No
DIC-1 level model	1645.62	1586.56	1646.82	1098.83
DIC-2 level model	1646.59	1586.08	1633.47	1101.18

Burn-in 5000; chain 200,000; Bayesian Deviance Information Criteria is used to examine for model fit improvement in single level compared to multilevel models; * *p*
<0.05 z-score test; motives are log-transformed.

**Table 3 ijerph-16-00853-t003:** Coping motives regressed on neighborhood and individual measures (95% credible intervals) n = 1046 (Intermediate Data Zones n = 188).

Predictor Variable	Model 1	Model 2	Model 3
*Demographics*			
Sex (male) Ref: female	−0.18 (−0.24, −0.12) ***	−0.18 (−0.24, −0.12) ***	−0.17 (−0.24, −0.11) ***
Family structure (Ref: both parents)			
single parent	0.09 (0.01, 0.17) *	0.08 (−0.00, 0.16)	0.07 (−0.01, 0.15)
other	0.04 (−0.05, 0.13)	0.04 (−0.06, 0.13)	0.04 (−0.06, 0.13)
Family Affluence (Ref: low)			
medium	−0.07 (−0.14, 0.01)	−0.04 (−0.12, 0.03)	−0.04 (−0.12, 0.04)
high	−0.06 (−0.13, 0.02)	−0.02 (−0.10, 0.05)	−0.02 (−0.09, 0.06)
*Neighborhood Conditions*			
On-trade license density		0.00 (−0.01, 0.01)	0.00 (−0.01, 0.01)
Off-trade license density		−0.01 (−0.04, 0.02)	−0.01 (−0.04, 0.01)
Urban/rurality (Ref: large cities)			
other urban		0.06 (−0.06, 0.19)	0.07 (−0.05, 0.20)
accessible small towns		0.14 (0.00, 0.28) *	0.14 (0.00, 0.28) *
accessible rural		0.08 (−0.05, 0.21)	0.08 (−0.05, 0.21)
remote small towns		0.11 (−0.03, 0.24)	0.11 (−0.02, 0.25)
remote rural		0.03 (−0.10, 0.16)	0.02 (−0.11, 0.15)
Neighborhood deprivation (Ref: 1 most deprived)			
2		−0.14 (−0.24, −0.05) **	−0.14 (−0.24, −0.04) **
3		−0.11 (−0.21, −0.01) *	−0.10 (−0.20, 0.00)
4 least deprived		−0.17 (−0.27, −0.06) **	−0.16 (−0.27, −0.06) **
Neighborhood-level social cohesion		0.05 (−0.13, 0.23)	0.10 (−0.10, 0.29)
Neighborhood-level disorder		0.39 (0.09, 0.68) *	0.26 (−0.06, 0.58)
*Perceptions*			
Perceived social cohesion			−0.01 (−0.02, 0.00)
Perceived disorder			0.03 (0.00, 0.05) *
Neighborhood variance	0.017 (0.005, 0.032)	0.014 (0.003, 0.028)	0.014 (0.003, 0.027)
Individual variance	0.231 (0.210, 0.254)	0.228 (0.207, 0.251)	0.226 (0.206, 0.249)
Bayesian DIC	1492.90	1482.52	1476.68
Residual Moran’s I			0.0190 (*p* = 0.449)

Burn-in 5,000 chain length 200,000; DIC = Deviance Information Criteria; * p<0.05, ** p<0.01, *** p<0.001; coping is log-transformed.

**Table 4 ijerph-16-00853-t004:** Unstandardized coefficients (standard errors) for path models.

	Perceived Disorder	Coping Motives	Weekly Drinking
*Demographics*			
Male	−0.14 (0.08)	0.16 (0.03) ***	0.32 (0.09) ***
Family structure (Ref: both parents)			
single-parent family	0.15 (0.11)	0.08 (0.04) *	0.21 (0.09) *
other	−0.06 (0.12)	0.04 (0.05)	0.03 (0.12)
*Neighborhood Conditions*			
Neighborhood deprivation (Ref: 1 most deprived)			
2		−0.14 (0.04) **	0.01 (0.12)
3		−0.09 (0.05)	−0.13 (0.12)
4 least deprived		−0.17 (0.05) **	−0.22 (0.13)
Urban/rurality (Ref: large cities)			
other urban		0.08 (0.07)	0.11 (0.16)
accessible small towns		0.15 (0.07) *	0.28 (0.15)
accessible rural		0.10 (0.07)	−0.05 (0.15)
remote small towns		0.11 (0.07)	−0.01 (0.16)
remote rural		0.06 (0.06)	0.11 (0.15)
Neighborhood-level disorder	5.09 (0.22) ***		
*Potential Mediators*			
Perceived disorder		0.03 (0.01) ***	0.06 (0.03)*
Coping motives			0.66 (0.07) ***

Fit statistics: estimated degrees of freedom = 39; CFI = 0.979; TLI = 0.920; RMSEA = 0.032; coping motives are log-transformed; * *p*
<0.05; ** p<0.01; *** p<0.01.

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
