# Peer review of "Do Drinking Motives Mediate the Relationship between Neighborhood Characteristics and Alcohol Use among Adolescents?"

_ijerph, 2019, doi:10.3390/ijerph16050853_

Round 1

Reviewer 1 Report

This paper examines the possibility of drinking motives mediating the relationship between neighbourhood characteristics and alcohol use in young people.  It’s a nice idea and on the whole the paper is well written, my primary concern is with the inclusion of family affluence as a covariate  as I think this means you are no longer testing what it is you set out to test. Specific comments below.

Abstract:

The wording “The study was limited to 1,119 . . . “ is a little weird for an abstract – why limited to?

Instruction:

The sentence on line 85 is very vague with little actual meaning – consider re-writing

Method:

Again, this “limited to” wording – what were the inclusion criterion, what number of people were excluded etc.

How does the item response model account for sex? Was it stratified?

The reliability for the neighbourhood level measures seems low, can you justify this?

I’m a bit confused by the inclusion of family affluence as a covariate – by doing this you are effectively testing to see if drinking motives mediate the gap between family and neighbourhood affluence rather than what you are saying that you are testing. I would personally, think the paper would be better with this removed. 

Sections of the analysis section are difficult to follow – what does burn-in mean?

Results

Please consider introducing the discussion of each table with a sentence explaining what the purpose of explaining that data is.

I could take or leave the minimum and maximum scores in Table 1.

Table 2: What is a 95% credible interval?

The title in Table 2 needs rewording – the information presented should stand alone and “Empty models examining drinking motives across neighbourhoods” does not give enough information.

Discussion

I think your conclusion does not match your results well – you discuss the link between neighbourhood inequality and drinking to cope which makes sense to me, but you then finish on a recommendation for a program that helps people to adapt to neighbourhood deprivation, rather than pointing out a potential side-effect of a reduction in neighbourhood deprivation.   

Author Response

Reviewer 1:

Reviewer: This paper examines the possibility of drinking motives mediating the relationship between neighbourhood characteristics and alcohol use in young people.  It’s a nice idea and on the whole the paper is well written, my primary concern is with the inclusion of family affluence as a covariate  as I think this means you are no longer testing what it is you set out to test. Specific comments below.

Response: We thank the reviewer for these comments. We feel the paper is improved in addressing your thoughtful suggestions. Responses to specific comments are below (including the observation regarding family affluence).

Reviewer: Abstract:

The wording “The study was limited to 1,119 . . . “ is a little weird for an abstract – why limited to?

Response: This is removed from the abstract and now reads “The study included 1119 . . . “

Reviewer: Instruction:

The sentence on line 85 is very vague with little actual meaning – consider re-writing

Response: This sentence is now changed to:

“Research is needed to identify sub-groups of young people who have varying drinking motives.”

Reviewer: Method:

Again, this “limited to” wording – what were the inclusion criterion, what number of people were excluded etc.

Response: this was changed from “limited to” to “included”

The total boosted sample had 3591 students (line 103).

To increase clarity the description of the sample now reads:

“To be included in the analysis students had to report their home postcode and reside in an IDZ that had 5 or more survey participants (n=1558). This allowed for reliable measures of neighbourhood-level social cohesion and disorder {Martin2017, Martin2018, Prins2014}. From there, students were included in the analysis only if they reported positively to having ever drank and to having consumed alcohol in the past 12 months (n=1119), as the drinking motives questions were only asked of current drinkers.  Those included in the study were significantly (p<0.05) more likely to be in the high family-affluence tertile (40 versus 34%) than those excluded but were not significantly different in terms of family structure and sex.”

Reviewer: How does the item response model account for sex? Was it stratified?

Response: Sex is included as an individual-level covariate in the item response model. More details are available from Martin et al. 2017.

Th following description was added to increase clarity.

“Neighbourhood-level social cohesion and neighbourhood-level disorder measures were constructed by combining individual responses within an IDZ unit. This was done using a three-level item response model (level 1=item; level 2=pupil; level 3= neighbourhood) accounting for item severity and pupil's sex as an individual-level covariate. More details can be found in {Martin2017}. The reliability at the neighbourhood-level for the social cohesion and disorder measures were 0.577 and 0.563, respectively. Previous research has considered similar neighbourhood-level reliability values acceptable {Prins2012}. “

Reviewer: The reliability for the neighbourhood level measures seems low, can you justify this?

Response: There is currently no defined cut-off for an acceptable level of reliability at the neighbourhood level. However, the measures reliabilites are consistent with a study of neighbourhood level social capital among Dutch adolescents, which was deemed adequate (Prins et al. 2012). We make note of this now in the manuscript.

“Previous research has considered similar neighbourhood-level reliability values acceptable  {Prins2012}.”

Reviewer: I’m a bit confused by the inclusion of family affluence as a covariate – by doing this you are effectively testing to see if drinking motives mediate the gap between family and neighbourhood affluence rather than what you are saying that you are testing. I would personally, think the paper would be better with this removed.

Response: We included the FAS in analyses as this allows for the association of neighbourhood -level deprivation (neighbourhood is the characteristic of interest) with drinking motives to be assessed independent of family level- affluence. Therefore, results can be interpreted as the effect of neighbourhood deprivation controlling for individual’s family affluence (context not composition).

We now specify this in the manuscript, so it is clear what the model is measuring. Ross et al. 2008 goes into this in detail, so we now include this citation in the manuscript.

“To test the influence of neighbourhood deprivation independent of family socio-economic status {Ross2008}, and because there is evidence that drinking to cope varies by family socio-economic condition {Stapinski2016}, family affluence was included as a covariate.”

Reviewer: Sections of the analysis section are difficult to follow – what does burn-in mean?

Response: We have now added more details to the analysis section, and several references:

“These analyses were conducted using runmlwin in Stata and MLWin using Markov chain Monte Carlo (MCMC). These procedures do not produce point estimates, instead many iterations are run and for each evaluation a distribution is formed. From this accuracy interval estimates are produced, namely credible intervals, which can be interpreted similarly to confidence intervals. The mean of the distribution can be obtained and used as substitute for a point estimate. A p value can be derived and interpreted as the probability of the (null) hypothesis {Vandeschoot2014}. A start value needs to be given for MCMC sampling. For this research the values were given using a least squares method {Leckie2013}. Generally, the model iterations take a while to `settle down' (converge) so some iterations are omitted from the sample from which the summary values are drawn. This is called the burn-in {Browne2017}. Like past research, a burn-in length of 5,000 was used {Declercq2014}. The number of iterations (chain length) needed to achieve model convergence was set at 200 000 based on the Raftery-Lewis statistic and visual examination of the trajectory plots {Declercq2014}(see Appendix Material). Bayesian Deviance Information Criteria (DIC) was used to test model fit across models with a smaller DIC indicating better fit and a difference of 5 considered substantial Khana2018, Spiegelhalter2002}.”

Reviewer: Results

Please consider introducing the discussion of each table with a sentence explaining what the purpose of explaining that data is.

Response: The following sentences were added:

Table 2 shows the results of the empty models.

Table 3 gives the results of multivariable models.

Reviewer: I could take or leave the minimum and maximum scores in Table 1.

Response: These columns were removed.

Reviewer: Table 2: What is a 95% credible interval?

Response: This is now addressed in the analysis section (see above).

Reviewer: The title in Table 2 needs rewording – the information presented should stand alone and “Empty models examining drinking motives across neighbourhoods” does not give enough information.

Response: This has been changed to - Empty models of variation in drinking motives across neighbourhoods (Intermediate Data Zones) (95% credible intervals)

Reviewer: Discussion

I think your conclusion does not match your results well – you discuss the link between neighbourhood inequality and drinking to cope which makes sense to me, but you then finish on a recommendation for a program that helps people to adapt to neighbourhood deprivation, rather than pointing out a potential side-effect of a reduction in neighbourhood deprivation. 

Response: We thank the reviewer for this observation.  We have altered one paragraph in the discussion to improve clarity and adjusted or conclusion:

“The link between deprivation and drinking to cope may be explained by two, not mutually exclusive, hypotheses: 1) deprived neighbourhoods create stress due to the physical and social conditions of the neighbourhood, and alcohol is used to cope with the stress created by the environment, and 2) those residing in such neighbourhoods have different strategies for coping with life's general stresses and are more likely to use alcohol to deal with problems {Karriker-Jaffe2016}. Because coping-motivated drinking may represent a form of self-medication {Stapinski2016}, our findings suggests the need to develop and evaluate strategies that can help individuals cope with negative affect without alcohol. Additionally, removing neighbourhood stress through programmes to improve neighbourhood conditions may reduce drinking to cope, as well as having wider benefits.”

“In conclusion, of the four motives examined, only coping motives varied across neighbourhoods. Based on these findings, public health policies that develop adaptive strategies to improve alcohol-free methods for young people to cope better with life's stresses may be particularly effective at reducing inequalities, if targeted at young people living in accessible small-towns or areas of high neighbourhood deprivation. Additionally, reduction of coping motives and regular alcohol use may be a potential side effect of improving neighbourhood conditions.”

Reviewer 2 Report

Introduction

The authors argue that there are currently no studies that examine the extent to which adolescent drinking motives vary across neighbourhoods, but then later in the text refer to few studies that already examined this research question (e.g. Portuguese study) à inconsistent statements should be changed

Materials and Methods

please already report some basic characteristics of participants in the sample section as age and gender distribution

Is it right that the interview data is from 2010, but data about alcohol outlet density is from 2012? Probably this has not changed a lot but the authors should address this point briefly

References Currie et al. 2009 and Currie et al. 2008 are not listed in the references

Is there any evidence, that the four questions (family vehicle, computer ownership, own bedroom, family holidays) are valid measures of family affluence? and how did the authors define low, medium and high family affluence?

Levin et al. 2014 should be changed into a numbered reference, Shortt et al. 2015 is not listed in the reference section

The authors explain how they measured “perceived disorder” but it remains unclear what exactly was defined as “neighbourhood disorders”. Furthermore I do not see the point in using a five point scale for perceived social cohesion but a three point scale for perceived disorders

what was the income in the deprivation quintiles?

Results:

 line 258: which table?

again, measures of “neighbourhood disorders” are unclear

Discussion:

ll 353 -355: please explain these study results and  their relation to your work more detailed

the interpretation about the link between deprivation and drinking to cope are somehow confusing. On the one hand, the authors interpret their findings as contradicting the hypothesis that  “alcohol is used to cope with the stress created by the environment”, but on the other, they also state their findings support similar findings of “… perceiving the local area as a more problematic neighbourhood gives rise to stronger drinking motives generally.”

the authors emphasize the “need for targeted strategies that can help individuals cope with negative affect without alcohol”, but they fail to give examples of these targeted strategies in different neighbourhoods

Author Response

Reviewer 2:

We thank the reviewer for their time in reviewing this manuscript. Their suggestions were very helpful, and we believe in addressing them we have improved the paper.

Introduction

The authors argue that there are currently no studies that examine the extent to which adolescent drinking motives vary across neighbourhoods, but then later in the text refer to few studies that already examined this research question (e.g. Portuguese study) à inconsistent statements should be changed

Response: The research conducted to date (i.e., the Portuguese study) does not consider geographic space or neighbourhood membership and therefore, does not examine for variation across neighbourhoods, but rather each individual’s perception of their neighbourhood. This is now clarified up front on line 29.

“Currently there are no studies that examine the extent to which adolescent drinking motives vary across neighbourhoods; existing studies have examined perceptions of neighbourhoods but ignore neighbourhood membership. Understanding if drinking motives are associated with where adolescents reside allows for a better comprehension of the pathways by which neighbourhood impacts adolescent alcohol use.”

This is also addressed on lines 80-81 that states “Although these studies suggest there may be an association between neighbourhood characteristics and adolescent drinking motives, they deal only with individual perceptions of neighbourhood…”

Reviewer: Materials and Methods

please already report some basic characteristics of participants in the sample section as age and gender distribution

Response: This is now included on lines 250-251.

Reviewer: Is it right that the interview data is from 2010, but data about alcohol outlet density is from 2012? Probably this has not changed a lot but the authors should address this point briefly

Response: The reviewer is correct. This is now addressed in the limitations section.

Reviewer: References Currie et al. 2009 and Currie et al. 2008 are not listed in the references

Response: This has been corrected.

Reviewer: Is there any evidence, that the four questions (family vehicle, computer ownership, own bedroom, family holidays) are valid measures of family affluence? and how did the authors define low, medium and high family affluence?

Response: The FAS has been validated and is widely used in Europe and North America.  It has been found to have better criterion validity and to be less affected by nonresponse bias than questions about parental education or income.  A citation that addresses this is added (Boyce et al. 2006). Low, medium, and high family affluence were defined by tertiles for the total sample of Scottish adolescents. An additional reference is added that provides more details on this approach (Batista-Foguet 2004).

Reviewer: Levin et al. 2014 should be changed into a numbered reference, Shortt et al. 2015 is not listed in the reference section

Response: This has been corrected.

Reviewer: The authors explain how they measured “perceived disorder” but it remains unclear what exactly was defined as “neighbourhood disorders”. Furthermore I do not see the point in using a five point scale for perceived social cohesion but a three point scale for perceived disorders

Response: Neighbourhood-level disorder is an aggregate of individual responses within an intermediate data zone (IDZ) unit, yielding a single value for each IDZ. This was done using an item response model. A full description of this method is found in Martin et al.  2017.

To improve clarity this now reads:

“Neighbourhood-level social cohesion and neighbourhood-level disorder measures were constructed by combining individual responses within an IDZ unit. This was done using a three-level item response model (level 1=item; level 2= pupil; level 3= neighbourhood) accounting for item severity and pupil's sex as an individual-level covariate. More details can be found in Martin2017.”

The items regarding disorder were to do with the quantity of features in a neighbourhood, and therefore are not directly equivalent to the social cohesion items which include statements regarding relationships in the neighbourhood. This is reflected in the different response options used.

Reviewer: what was the income in the deprivation quintiles?

Response: the income domain of the SIMD is comprised of several income indicators. It is a measure of adults and their dependents in receipt of Income Support, Employment and Support Allowance, Job Seekers Allowance, Guaranteed Pension Credits, and Child and Working Tax Credits.

This now reads “Neighbourhood deprivation was determined by the income domain of the Scottish Index of Multiple Deprivation (SIMD) 2012, which is comprised of several indicators of income deprivation (Scottish Government 2012)”

Reviewer: Results:

Reviewer: line 258: which table?

Response: This is corrected

Reviewer: again, measures of “neighbourhood disorders” are unclear

Response: we have changed the wording of neighbourhood disorder to neighbourhood-level disorder or perceived disorder throughout to clarify. The same was done for social cohesion for consistency.

Reviewer: Discussion:

Reviewer: ll 353 -355: please explain these study results and  their relation to your work more detailed

Response: We have now added the sentence to tie our research to previous findings “Therefore, based on our findings, Scottish adolescents residing in more deprived areas or accessible small-towns may be at greater risk of alcohol related harms which has implications for their immediate and longer-term health.”

Reviewer: the interpretation about the link between deprivation and drinking to cope are somehow confusing. On the one hand, the authors interpret their findings as contradicting the hypothesis that  “alcohol is used to cope with the stress created by the environment”, but on the other, they also state their findings support similar findings of “… perceiving the local area as a more problematic neighbourhood gives rise to stronger drinking motives generally.”

Response: In accordance with this comment and comments by reviewer 1 this paragraph was changed to improve clarity .

“The link between deprivation and drinking to cope may be explained by two, not mutually exclusive, hypotheses: 1) deprived neighbourhoods create stress due to the physical and social conditions of the neighbourhood, and alcohol is used to cope with the stress created by the environment, and 2) those residing in such neighbourhoods have different strategies for coping with life's general stresses and are more likely to use alcohol to deal with problems {Karriker-Jaffe2016}. Because coping-motivated drinking may represent a form of self-medication {Stapinski2016}, our findings suggest the need to develop and evaluate strategies that can help individuals cope with negative affect without alcohol. Additionally, removing neighbourhood stress through programmes to improve neighbourhood conditions may reduce drinking to cope, as well as having wider benefits.”

Reviewer: the authors emphasize the “need for targeted strategies that can help individuals cope with negative affect without alcohol”, but they fail to give examples of these targeted strategies in different neighbourhoods

Response: We are hesitant to endorse any particular intervention, as there is a dearth of such programs that have been well evaluated. Rather we have now changed the language to endorse a need to design and evaluate such strategies.

“…our findings suggest the need to develop and evaluate strategies that can help individuals cope with negative affect without alcohol. “